# LEARNING TO COMPOSE SOFT PROMPTS FOR COMPOSITIONAL ZERO-SHOT LEARNING

**Nihal V. Nayak**[*]**, Peilin Yu**[*]**, Stephen H. Bach**
Department of Computer Science
Brown University
Providence, RI 02906, USA
`{nnayak2, pyu12, sbach}@cs.brown.edu`

## ABSTRACT

We introduce compositional soft prompting ($\mathbb{CSP}$), a parameter-efficient learning technique to improve the zero-shot compositionality of large-scale pretrained vision-language models (VLMs) like CLIP. We develop $\mathbb{CSP}$ for compositional zero-shot learning, the task of predicting unseen attribute-object compositions (e.g., old cat and young tiger). VLMs have a flexible text encoder that can represent arbitrary classes as natural language prompts but they often underperform task-specific architectures on the compositional zero-shot benchmark datasets. $\mathbb{CSP}$ treats the attributes and objects that define classes as learnable tokens of vocabulary. During training, the vocabulary is tuned to recognize classes that compose tokens in multiple ways (e.g., old cat and white cat). At test time, we recompose the learned attribute-object vocabulary in new combinations to recognize novel classes. We show that $\mathbb{CSP}$ outperforms the CLIP on benchmark datasets by an average of 10.9 percentage points on AUC. $\mathbb{CSP}$ also outperforms CoOp, a soft prompting method that fine-tunes the prefix context tokens, by an average of 5.8 percentage points on AUC. We perform additional experiments to show that $\mathbb{CSP}$ improves generalization to higher-order attribute-attribute-object compositions (e.g., old white cat) and combinations of pretrained attributes and fine-tuned objects. The code is available at https://github.com/BatsResearch/csp.

## 1 INTRODUCTION

Compositionality is the long-standing goal of artificial intelligence of creating new concepts by combining existing primitive concepts (Chomsky, 1956; Fodor & Pylyshyn, 1988; Hupkes et al., 2020; Lake & Baroni, 2018; Marcus, 2003). The practical advantage of compositionality for deep neural networks lies in the ability to build new classifiers by combining existing classifiers. In this work, we consider compositional zero-shot learning, a classification task where the model learns to predict unseen or novel compositions of primitive concepts (Naeem et al., 2021; Nagarajan & Grauman, 2018; Purushwalkam et al., 2019). Research on compositional zero-shot learning in language and vision focuses on attribute-object compositions such as `old tiger` and `young tiger`, where `tiger` is the object category described by the attributes `old` and `young`.

Existing methods for compositional zero-shot learning typically map attributes and objects to pretrained word embeddings and use a pretrained image encoder backbone to jointly align the image and the attribute-object text representations to learn compositionality (Li et al., 2020; Mancini et al., 2021a;b; Misra et al., 2017; Naeem et al., 2021; Nagarajan & Grauman, 2018; Purushwalkam et al., 2019; Xu et al., 2021). However, the pretraining of the word embeddings and image encoder is disjoint and isolated from each other, i.e., these methods learn to align image and text representations from scratch. These task-specific architectures also are limited in flexibility. For example, to adapt these methods to higher-order compositions with multiple attributes and objects such as `small furry cat` or `old white tiger`, the original architecture needs to be modified. The ability to generalize beyond the original training length is a key test for compositionality (Hupkes et al., 2020).

---

[*]Equal contribution

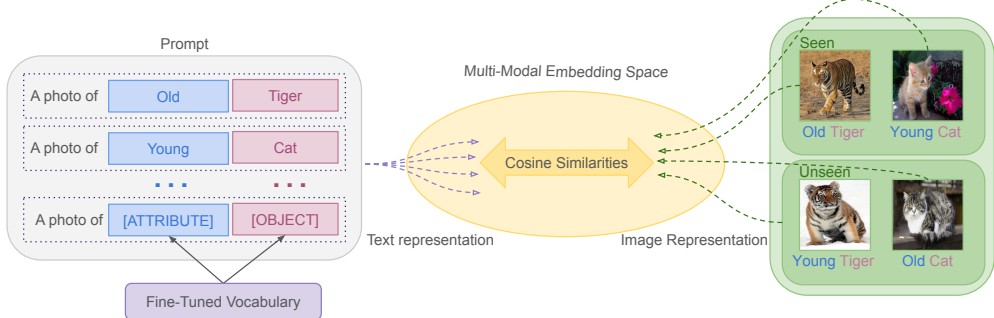

Figure 1: An overview of compositional zero-shot learning with ℂ𝕊ℙ. We fine-tune the vocabulary for attributes and objects on the seen classes. Then we compose novel soft prompts to test on the unseen classes.

In contrast, we propose to build on large-scale pretrained vision-language models (VLMs), which are trained on massive amounts of aligned images and text (Jain et al., 2021; Jia et al., 2021; Li et al., 2021; Radford et al., 2021). We focus on CLIP (Radford et al., 2021), a powerful vision-language model pretrained on 400 million image-text pairs. CLIP has two main components: the image encoder and the text encoder that produce vector representations for images and text in a multi-modal embedding space. The text encoder accepts a textual input, or a prompt such as `A photo of dog` to produce a vector representation for the class `dog`. Taking the cosine similarity with all the class prompts and the image, we get a compatibility score for the classes and pick the one with the highest score. However, CLIP without any fine-tuning underperforms task-specific architectures, even though it has been pre-trained on vastly more data. (See Appendix A for details.) This finding suggests that there is significant room for improvement from teaching VLMs like CLIP about composing concepts.

To improve VLMs for compositional zero-shot learning, we introduce compositional soft prompting (ℂ𝕊ℙ), a parameter-efficient learning technique that tunes tokens of vocabulary to represent primitive concepts in a composable way. Fine-tuning large pre-trained models such as CLIP requires huge amounts of compute and may lead to overfitting (Sung et al., 2021; Mitchell et al., 2022) (see also Section 5). This challenge has motivated several soft prompting techniques in both language and vision (Lester et al., 2021; Qin & Eisner, 2021; Vu et al., 2021; Zhou et al., 2021). These works tune a single prompt on a downstream supervised task, often in a few-shot setting. For instance, they typically use prompts such as `A photo of [class]` and tune the prefix `A photo of` on the entire dataset. In contrast, ℂ𝕊ℙ is a novel way of soft prompting. We treat the attributes and objects that are composed to define classes as learnable tokens of vocabulary in a prompt as `A photo of [attribute] [object]`. We tune on multiple `[attribute]` and `[object]` prompt compositions, and then we recompose them into new prompts for zero-shot inference (Figure 1).

Our results show that ℂ𝕊ℙ improves over the zero-shot performance of CLIP. ℂ𝕊ℙ significantly improves over CLIP across three benchmark datasets by an average accuracy of 13.7 percentage points in the closed-world setting and 8.0 percentage points in the open-world setting (using the AUC metric). ℂ𝕊ℙ also outperforms CoOp, a soft prompting method that tunes the prefix context, by an average of 7.3 percentage points in the closed-world setting and 4.3 percentage points in the open-world setting on the AUC metric.

In addition to improved benchmark accuracy, ℂ𝕊ℙ has several other advantages when tested on other kinds of zero-shot inferences without any changes to training. We show that the learned attribute vocabulary can be decomposed to better classify attributes in isolation, using prompts of the form `A photo of [attribute] object`. We also show that training ℂ𝕊ℙ with attribute-object compositions improves CLIP's performance on attribute-attribute-object compositions. Finally, we show that ℂ𝕊ℙ improves generalization to compositions of *unseen* attributes and seen objects. Prior work on compositional zero-shot learning typically only evaluates unseen compositions of seen attributes and seen objects.

In summary, our main contributions are:

1. We introduce compositional soft prompting ($\mathbb{CSP}$), a parameter-efficient learning technique to improve the compositionality of large-scale vision-language models (VLMs). The attributes and objects that are composed to define classes are treated as learnable tokens of vocabulary. Unlike existing work on soft prompting, our learned prompts are tuned on multiple compositions and then recomposed in new combinations for inference.
2. $\mathbb{CSP}$ improves the AUC accuracy of CLIP by an average of 10.9 percentage points across three benchmark datasets (Isola et al., 2015; Yu & Grauman, 2014; Mancini et al., 2021a). It also improves over CoOp, a soft-prompting method that tunes the prefix context, by an average of 5.8 percentage points on AUC.
3. We conduct additional experiments to analyze $\mathbb{CSP}$. We show that training on attribute-object compositions improves CLIP's accuracy on attribute classification alone, attribute-attribute-object compositions, and compositions of pretrained and fine-tuned vocabulary.

## 2 RELATED WORK

We describe the related work in prompting, parameter-efficient learning, and compositional zero-shot learning.

**Prompting**    Prompting is a recent focus in the language and vision communities that has shown benefits in zero-shot and few-shot learning on a wide range of tasks (Bach et al., 2022; Bommasani et al., 2021; Brown et al., 2020; Lester et al., 2021; Radford et al., 2021; Qin & Eisner, 2021; Sanh et al., 2022; Vu et al., 2021; Zhou et al., 2021; 2022). Discrete prompts are typically hand-written text inputs that provide guidelines to large pre-trained models such as CLIP, GPT-3 (Brown et al., 2020), etc. for inference without updating the model parameters. While manually engineering prompts can help achieve better accuracy, it is often time-consuming and impractical to find the best prompt.

Soft prompting is an alternative to discrete prompts, where a part of the prompt is learned by backpropagating without fine-tuning the entire model. Several works using soft prompts show improved accuracy compared to hand-crafted prompts (Lester et al., 2021; Li & Liang, 2021; Qin & Eisner, 2021; Shin et al., 2020; Vu et al., 2021; Zhou et al., 2021). In all these works, soft prompts are a single input concatenated to all inputs for the entire task. In contrast, we learn tokens for each primitive concept from multiple compositions and recompose them in new ways to represent unseen classes for zero-shot inference. We show in Section 5 that traditional soft prompting can also improve CLIP on compositional zero-shot learning, but generally not as much as $\mathbb{CSP}$.

Recent works have used soft prompting for large-scale vision-language models. Zhou et al. (2021) propose CoOp (contextual optimization), a soft prompting method for few-shot object classification. Other applications include visual question answering (Jin et al., 2022) and video understanding (Ju et al., 2021). Again, these works tune a single soft prompt on the entire dataset. One exception is Ge et al. (2022), which learns multiple soft prompts for cross-domain adaption. While our work shares similarities with their work, there are important differences. Our work decomposes the class labels into multiple parts rather than splitting the prompt into domain-related granularities such as domain-agnostic context, domain-specific context, and class label. Furthermore, we focus on compositional zero-shot learning where we do not have access to labeled examples from the unseen classes in the test set whereas they assume access to all the test classes during training.

Zhou et al. (2022) extend CoOp to CoCoOp (conditional contextual optimization) to reduce overfitting in prompt tuning for few-shot object classification. They learn a lightweight network that takes the image representation and produces a conditional vector that is added to the soft tokens in the prompt. In Appendix B, we compare CoCoOp and the analogous conditional variant of $\mathbb{CSP}$, CoCSP. While CoCSP improves over CoCoOp across three datasets in the AUC metric, CoCSP does not improve over $\mathbb{CSP}$ on these tasks.

**Parameter-Efficient Learning**    Soft prompting is closely related to the growing body of work in parameter-efficient learning (Houlsby et al., 2019; Guo et al., 2021; Mahabadi et al., 2021; Sung et al., 2021; Ben-Zaken et al., 2022; Liu et al., 2022). They add small feedforward networks between the layers in the pretrained model, or use sophisticated techniques to select a sparse set of model parameters and update them by fine-tuning on a labeled training set. However, unlike $\mathbb{CSP}$, the methods do not assign semantic meaning to the parameters in order to enable composition. In Appendix C, we experiment with CLIP adapters (Gao et al., 2021) for compositional zero-shot

Figure 2: Comparison of CLIP in a zero-shot and $\mathbb{CSP}$ in a compositional zero-shot setting.

learning. We find that this approach, which was designed for few-shot learning, reduces accuracy on unseen classes in three benchmark tasks.

**Compositional Zero-Shot Learning**    The growing interest in compositional zero-shot learning has contributed to several architectural innovations (Li et al., 2020; Mancini et al., 2021a;b; Misra et al., 2017; Naeem et al., 2021; Nagarajan & Grauman, 2018; Purushwalkam et al., 2019; Radenović et al., 2021). Early works compose attributes and objects with a transformation function (Misra et al., 2017; Nagarajan & Grauman, 2018). Recent work uses separate encoding layers for attributes and objects, and then combines them with late fusion using a linear layer or a multilayer perceptron (Purushwalkam et al., 2019). The most successful methods represent the attribute and object relationship in a graph and learn their compositions via graph convolutional networks (Mancini et al., 2021b; Naeem et al., 2021; Ruis et al., 2021). Yun et al. (2022) study the emergence of compositionality in CLIP pre-training by learning linear classifiers rather than soft prompts for primitive concepts. Scialom et al. (2022) show that continually fine-tuning large language models can learn composable instructions.

Evaluating CLIP in a fair setting compared to the existing task-specific architectures for compositional zero-shot learning is challenging. On the one hand, CLIP is trained on a web-scale dataset to which other methods do not have access, and may even contain some of the classes from the unseen split. On the other hand, existing task-specific architectures fine-tune orders of magnitude more parameters than soft prompting, while using specialized architectures that do not adapt as easily as VLMs to new tasks. For these reasons, our work focuses on improving CLIP-based baselines, and we include a summary of the results comparing task-specific architectures in Section 5 and extended results in Appendix A.

Compositional zero-shot learning is also closely related to the broader goal of compositionality in artificial intelligence (Chomsky, 1956; Fodor & Pylyshyn, 1988). Existing works have studied compositionality for image generation (Herzig et al., 2020), video synthesis (Ye et al., 2019; Bar et al., 2021; Nawhal et al., 2020), visual reasoning (Johnson et al., 2017), semantic parsing (Drozdov et al., 2022), language grounding (Jin et al., 2022), and question answering (Yuan et al., 2019). For more in-depth discussion on compositionality, we refer the readers to Hupkes et al. (2020).

## 3    PRELIMINARIES

In this section, we formally define the task of compositional zero-shot learning and describe how to use CLIP for compositional zero-shot inference. Let $\mathbb{A} = \{a_0, a_1, ..., a_n\}$ be the set of possible attributes and $\mathbb{O} = \{o_0, o_1, ..., o_m\}$ be the set of possible object categories. Let the label space $\mathbb{Y}$ be the Cartesian product of the attribute set and the object category set, $\mathbb{Y} = \mathbb{A} \times \mathbb{O}$. We are given two disjoint label subsets such that $\mathbb{Y}_{seen} \subset \mathbb{Y}$, $\mathbb{Y}_{unseen} \subset \mathbb{Y}$, and $\mathbb{Y}_{seen} \cap \mathbb{Y}_{unseen} = \varnothing$ where $\mathbb{Y}_{seen}$ and $\mathbb{Y}_{unseen}$ are the set of the seen and unseen classes. At training time, we are given examples $\mathbb{S}_{seen} = \{(x_1, y_1), ..., (x_n, y_n)\}$ to learn some discriminative model $f : \mathbb{X} \to \mathbb{Y}_{seen}$. During inference, we want the model to predict both seen and unseen classes in the test set, i.e., $f : \mathbb{X} \to \mathbb{Y}_{test}$. In the closed-world evaluation, the test set is defined as $\mathbb{Y}_{test} = \mathbb{Y}_{seen} \cup \mathbb{Y}_{unseen}$. In the open-world evaluation, the model has to consider all possible permutations of the attribute-object compositions, i.e., $\mathbb{Y}_{test} = \mathbb{Y}$ and $\mathbb{Y}_{unseen} = \mathbb{Y} - \mathbb{Y}_{seen}$.

We can easily adapt CLIP to compositional zero-shot inference by defining prompts appropriately. The prompts used in CLIP for zero-shot inference differ from the prompts used in works like GPT-3 (Brown et al., 2020) and T0 (Sanh et al., 2022). Instead of defining one prompt for an $N$-way classification task, the $N$ classes are transformed into natural language prompts such as `A photo of [class]` (Figure 2, left). This results in $N$ prompt vectors, which are used to compute the cosine similarity with the image representation for prediction. For compositional zero-shot inference, we

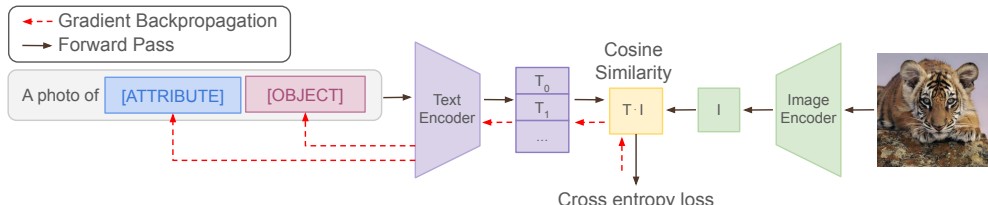

Figure 3: Training setup for $\mathbb{CSP}$. The prompt with the attribute and object vocabulary is passed through the text encoder to get the text representation. The example is passed through the image encoder for the image representation. Next, we take the cosine similarity for all the prompts with the image and compute the cross entropy loss. Finally, we backpropagate the loss through the text encoder and update the attribute and object vocabulary weights.

replace the `[class]` placeholder with the attribute and object composition as follows: `a photo of [attribute] [object]`. The change in prompt format allows us to represent pairs of attribute and objects such as `a photo of young cat` in the text encoder without changes.

## 4 COMPOSITIONAL SOFT PROMPTING

In this section, we introduce compositional soft prompting ($\mathbb{CSP}$), a parameter-efficient learning technique for fine-tuning large pretrained models for better compositionality.

**Motivation**    The goal of our work is to improve VLMs such as CLIP on compositional generalization where they seem to underperform the current state-of-the-art methods (Appendix A). This is perhaps because CLIP's pretraining on data crawled from the web does not provide sufficient supervision about attributes and how they combine with different objects. Therefore, we aim to teach VLMs such as CLIP how to better compose primitive concepts. We approach this is a vocabulary-learning problem because it is parameter efficient and gives us a natural way to compose new classes.

**Prompt Construction**    $\mathbb{CSP}$ treats the attributes and objects that are composed to define classes as learnable tokens of vocabulary and tunes them on multiple prompt compositions. We represent each primitive concept, either attribute or object, as a new, auxiliary token in the VLM's vocabulary. To represent a class, we combine a fixed prefix and the corresponding primitive concepts, for example `A photo of young tiger`, where `young` maps to the auxiliary weight vector for the corresponding attribute and `tiger` maps to the auxiliary weight vector for the correponding object. Then, we use these prompts for compositional zero-shot learning in the same way as we would for CLIP, as described in Section 3. Training and inference with $\mathbb{CSP}$ is very simple as we only need to swap the vocabulary of the attributes and objects for any desired composition in the prompt (Figure 2, right). As a result, our method tunes only $(|\mathbb{A}| + |\mathbb{O}|) \times d$ parameters where $d$ is the dimension of the vocabulary embedding. This is a novel form of soft prompting, because prior work (Lester et al., 2021; Qin & Eisner, 2021; Zhou et al., 2021) tune prompts for seen classes and usually one prompt for the whole task, whereas we compose learned prompt tokens to represent unseen classes at test time.

**Training**    We want to learn vector embeddings for the new, auxiliary vocabulary: $\boldsymbol{\theta} = [\boldsymbol{\theta}_{\mathbb{A}}; \boldsymbol{\theta}_{\mathbb{O}}]$ where $\boldsymbol{\theta} \in \mathbb{R}^{(|\mathbb{A}|+|\mathbb{O}|) \times d}$. We learn them by composing prompts with them and fine-tuning them on the seen classes. Figure 3 shows the overall learning process for $\mathbb{CSP}$.

First, we initialize the learnable vocabulary with pretrained embeddings from CLIP using the concept names, i.e. attributes and objects. If a concept, such as `Faux Fur`, has multiple tokens, we average their pre-trained representations to initialize the learnable vocabulary. Next, for each class, we construct a prompt with the pretrained vocabulary for the prefix context and learnable vocabulary for the primitive concepts:

$$t_{a,o} = (\boldsymbol{w}_0, \ldots, \boldsymbol{w}_p, \boldsymbol{\theta}_a, \boldsymbol{\theta}_o)$$

where $\boldsymbol{w}_i \in \mathbb{R}^d$ are the tokens of the prefix context, and $\boldsymbol{\theta}_a$ and $\boldsymbol{\theta}_o$ are the learnable parameters for the attribute and the object in the prompt.

| Dataset | Composition | | | Train | | Validation | | | Test | | |
|---|---|---|---|---|---|---|---|---|---|---|---|
| | $|\mathbb{A}|$ | $|\mathbb{O}|$ | $|\mathbb{A} \times \mathbb{O}|$ | $|\mathbb{Y}_{seen}|$ | $|\mathbb{X}|$ | $|\mathbb{Y}_{seen}|$ | $|\mathbb{Y}_{unseen}|$ | $|\mathbb{X}|$ | $|\mathbb{Y}_{seen}|$ | $|\mathbb{Y}_{unseen}|$ | $|\mathbb{X}|$ |
| MIT-States Isola et al. (2015) | 115 | 245 | 28175 | 1262 | 30338 | 300 | 300 | 10420 | 400 | 400 | 12995 |
| UT-Zappos Yu & Grauman (2014) | 16 | 12 | 192 | 83 | 22998 | 15 | 15 | 3214 | 18 | 18 | 2914 |
| C-GQA Mancini et al. (2021a) | 413 | 674 | 278362 | 5592 | 26920 | 1252 | 1040 | 7280 | 888 | 923 | 5098 |

Table 1: Summary statistics of the datasets used in our experiments.

Next, we pass the prompt with learnable parameters through the text encoder to get the text representation:

$$\boldsymbol{t}_{a,o} = \frac{VLM_{\mathbf{T}}\left(t_{a,o}\right)}{||VLM_{\mathbf{T}}\left(t_{a,o}\right)||}$$

Then, for some image $x$, we get the image representation from the image encoder:

$$\boldsymbol{x} = \frac{VLM_{\mathbf{V}}\left(x\right)}{||VLM_{\mathbf{V}}\left(x\right)||}$$

Finally, we compute the class probability:

$$p_{\boldsymbol{\theta}}(y = (a, o) \mid x) = \frac{\exp\left(\boldsymbol{x} \cdot \boldsymbol{t}_{a,o}/\tau\right)}{\sum_{(\hat{a},\hat{o}) \in \mathbb{Y}_{seen}} \exp\left(\boldsymbol{x} \cdot \boldsymbol{t}_{\hat{a},\hat{o}}/\tau\right)}$$

where $\tau \in \mathbb{R}$ is the temperature parameter from CLIP. We learn the parameters $\boldsymbol{\theta}$ by minimizing the cross entropy loss on the training dataset:

$$-\frac{1}{|\mathbb{S}_{seen}|} \sum_{(x,y) \in \mathbb{S}_{seen}} \log p_{\boldsymbol{\theta}}(y \mid x) + \lambda||\boldsymbol{\theta}||^2$$

where $\lambda \in \mathbb{R}$ is the weight decay.

**Inference** During inference, we recompose the fine-tuned attribute and object vocabulary in the prompt. We compose the candidate prompts with the tuned $\boldsymbol{\theta}$ with the (attribute, object) pairs in the same way during training. In both closed-world and open-world settings, we only replace attribute and objects with the fine-tuned parameters in the prompt. Finally, we calculate the most likely attribute and object pair as follows:

$$\hat{y} = \underset{y \in \mathbb{Y}_{test}}{\arg\max} \ p_{\boldsymbol{\theta}}(y \mid x)$$

We include the pseudocode for inference in Appendix F.

## 5 EXPERIMENTAL EVALUATION

In this section, we describe our experiments with $\mathbb{CSP}$. We compare $\mathbb{CSP}$ to CLIP-based baselines in the closed-world and open-world settings of compositional zero-shot learning. We also compare the performance of fine-tuned CLIP with $\mathbb{CSP}$ on the benchmark datasets. Finally, we demonstrate that $\mathbb{CSP}$ can generalize beyond these benchmarks to three modified settings: attribute-only classification, attribute-attribute-object composition, and inference with unseen attributes.

**Dataset** We experiment with three attribute-object composition benchmarks: MIT-states (Isola et al., 2015), UT-Zappos (Yu & Grauman, 2014), and C-GQA (Naeem et al., 2021). Table 1 summarizes the statistics of the datasets. MIT-states contains images of naturally occurring objects where each object is described by an adjective. UT-Zappos contains images of shoes paired with fine-grained states. For this dataset, we use the split suggested by Purushwalkam et al. (2019). C-GQA, a newly introduced dataset derived from the Stanford GQA dataset (Hudson & Manning, 2019), contains images of objects paired with states.

**Benchmark Evaluation** We follow the standard closed-world and open-world evaluation protocols. In the closed-world setting, the unseen classes are a subset of all possible attribute-object combinations and are defined in the dataset. In the open-world setting, the model considers all possible attribute-object combinations.

| | Method | MIT-States | | | | UT-Zappos | | | | C-GQA | | | |
|---|---|---|---|---|---|---|---|---|---|---|---|---|---|
| | | S | U | H | AUC | S | U | H | AUC | S | U | H | AUC |
| Closed | CLIP | 30.2 | 46.0 | 26.1 | 11.0 | 15.8 | 49.1 | 15.6 | 5.0 | 7.5 | 25.0 | 8.6 | 1.4 |
| | CoOp | $34.4_{0.1}$ | $47.6_{0.1}$ | $29.8_{0.1}$ | $13.5_{0.0}$ | $52.1_{0.5}$ | $49.3_{1.8}$ | $34.6_{1.7}$ | $18.8_{1.4}$ | $20.5_{0.2}$ | $\mathbf{26.8}_{0.3}$ | $17.1_{0.2}$ | $4.4_{0.1}$ |
| | $\mathbb{CSP}$ | $\mathbf{46.6}_{0.1}$ | $\mathbf{49.9}_{0.1}$ | $\mathbf{36.3}_{0.1}$ | $\mathbf{19.4}_{0.1}$ | $\mathbf{64.2}_{0.7}$ | $\mathbf{66.2}_{1.2}$ | $\mathbf{46.6}_{1.2}$ | $\mathbf{33.0}_{1.3}$ | $\mathbf{28.8}_{0.1}$ | $\mathbf{26.8}_{0.1}$ | $\mathbf{20.5}_{0.1}$ | $\mathbf{6.2}_{0.0}$ |
| Open | CLIP | 30.1 | 14.3 | 12.8 | 3.0 | 15.7 | 20.6 | 11.2 | 2.2 | 7.5 | 4.6 | 4.0 | 0.27 |
| | CoOp | $34.6_{0.1}$ | $9.3_{0.0}$ | $12.3_{0.1}$ | $2.8_{0.0}$ | $52.1_{0.5}$ | $31.5_{2.9}$ | $28.9_{2.3}$ | $13.2_{1.6}$ | $21.0_{0.2}$ | $4.6_{0.1}$ | $5.5_{0.1}$ | $0.70_{0.0}$ |
| | $\mathbb{CSP}$ | $\mathbf{46.3}_{0.3}$ | $\mathbf{15.7}_{0.1}$ | $\mathbf{17.4}_{0.1}$ | $\mathbf{5.7}_{0.0}$ | $\mathbf{64.1}_{0.7}$ | $\mathbf{44.1}_{0.3}$ | $\mathbf{38.9}_{0.5}$ | $\mathbf{22.7}_{0.4}$ | $\mathbf{28.7}_{0.2}$ | $\mathbf{5.2}_{0.1}$ | $\mathbf{6.9}_{0.1}$ | $\mathbf{1.20}_{0.0}$ |

Table 2: Closed-world (Closed) and open-world (Open) results on MIT-States, UT-Zappos, and C-GQA. For CoOp and $\mathbb{CSP}$, we report the average performance of the models on 5 random seeds with standard error.

Following prior work (Mancini et al., 2021a), we report the performance in the generalized zero-shot learning for both the closed-world and the open-world settings. In generalized zero-shot learning, we include both the seen and unseen classes in the test set. Several works have noted that zero-shot models are biased towards the seen classes, so we follow the standard of adding a scalar bias to the unseen classes (Chao et al., 2016; Ruis et al., 2021) . Following prior work, we vary the bias from $-\infty$ to $+\infty$ to get a curve indicating the seen accuracy on the x-axis and unseen accuracy on the y-axis. We report the area under the curve (AUC) and select the operating point with the best harmonic mean (H) between the seen and unseen accuracy. We also report the best seen accuracy (S) when bias is $-\infty$ and the best unseen accuracy (U) when the bias is $+\infty$. We average over five random seeds, selecting the best-performing model after each epoch on the validation data.

Following prior work, we apply feasibility calibration to all methods for prediction in the open-world setting. The open-world setting is particularly challenging as the label space contains all possible combinations of attributes and objects in the dataset. For instance, the label space contains feasible compositions such as `young cat` and `eroded cliff` and infeasible compositions such as `eroded cat`. Existing work shows a significant drop in model performance from the closed-world setting to the open-world setting (Mancini et al., 2021a;b). As suggested in Mancini et al. (2021a), we apply a simple feasibility calibration based on GloVe embeddings (Pennington et al., 2014) to filter out infeasible compositions. For more details, see Appendix G.

**Baselines** In our experiments, we primarily compare with CLIP-based baselines. We compare $\mathbb{CSP}$ to pretrained CLIP (Radford et al., 2021) and CoOp (Zhou et al., 2021). CoOp is a soft-prompting method that learns the prefix context with limited labeled examples in a few-shot setting. CoOp resembles prompt tuning (Lester et al., 2021) applied to VLMs.

**Training Details** We implement $\mathbb{CSP}$ and the baselines with a pretrained CLIP model in PyTorch (Paszke et al., 2019). We use the CLIP model ViT-L/14 which is the largest available model in our experiments. Nonetheless, our method is agnostic to the choice of CLIP architecture. The pretrained CLIP ViT-L/14 model has a vision transformer (ViT) as the image encoder and a transformer as the text encoder.

We train $\mathbb{CSP}$ and CoOp by minimizing the cross entropy loss with the Adam optimizer over the seen split in the dataset for 20 epochs. We use a single NVIDIA RTX 3090 or V100 GPU depending on their availability to train all our models. For each dataset, we choose the best hyperparameters based on the performance on the validation split. For more details, refer to Appendix H.

**Benchmark Results** Our results in Table 2 show that $\mathbb{CSP}$ significantly improves over CLIP on all the benchmark datasets in the closed-world and open-world settings. In the closed-world setting, we outperform CLIP in the AUC metric by 8.4 points on MIT-states, 28.0 point on UT-Zappos, and 4.8 points on C-GQA. Additionally, we show that $\mathbb{CSP}$ beats CoOp, a soft-prompting method, in the AUC metric by 5.9 points on MIT-States, 14.2 points on UT-Zappos, and 1.8 points on C-GQA. In results in the open-world setting show that $\mathbb{CSP}$ improves over CLIP by 2.7 points on MIT-States, 20.5 points on UT-Zappos, and 0.93 points on C-GQA in the AUC metric. We outperform CoOp on MIT-States by 3.1 points, UT-Zappos by 9.5 points, and C-GQA by 0.50 points in the AUC metric.

We also report additional results on these benchmarks in the appendices. We compare CLIP-based methods to existing compositional zero-shot learning methods in Appendix A. $\mathbb{CSP}$ outperforms all these methods on two out of the three benchmarks on the AUC and harmonic mean metrics in open- and closed-world settings, but as discussed in Section 2 such comparisons come with several caveats, so we only include them in the appendices for additional context. We experiment with

different backbones in Appendix E and the trend is consistent with the results reports here. Finally, we qualitatively evaluate $\mathbb{CSP}$ in compositional zero-shot image to text retrieval in Appendix I.

**Comparison with Full Fine-Tuning**   We aim to understand the potential benefits of fine-tuning all the parameters of CLIP instead of a small number. To that end, we fine-tune all the parameters in CLIP on the seen classes in our benchmark datasets.

Table 3 shows that fine-tuning CLIP can improve generalization to unseen classes but still achieves lower AUC compared to $\mathbb{CSP}$ on UT-Zappos. In addition, fine-tuning CLIP requires GPUs with more memory and often meticulous hyperparameter selection (Dong et al., 2022).

| Method | MIT-States | UT-Zappos | C-GQA |
|---|---|---|---|
| CLIP | 11.0 | 5.0 | 1.4 |
| CLIP (FT) | $22.2_{0.1}$ | $24.5_{1.6}$ | $10.5_{0.2}$ |
| $\mathbb{CSP}$ | $19.4_{0.1}$ | $33.0_{1.3}$ | $6.2_{0.0}$ |

Table 3: Closed-world results comparing $\mathbb{CSP}$ and fine-tuned CLIP (FT). For $\mathbb{CSP}$ and CLIP (FT), we report the average AUC on 5 random seeds with standard error.

**Comparison with Task-Specific Architectures**   We compare $\mathbb{CSP}$ to existing task-specific compositional zero-shot learning architectures. We consider the following task-specific architectures: CGE (Naeem et al., 2021), Co-CGE (Mancini et al., 2021b) and ProtoProp (Ruis et al., 2021). These methods are the most recent best-performing methods in the closed-world setting.

Our results in Table 4 show that $\mathbb{CSP}$ outperform existing task-specific architectures on MIT-States and C-GQA while being competitive on UT-Zappos in AUC. We see that $\mathbb{CSP}$ improves over existing task-specific architectures by 12.8 points on MIT-states and 2.0 points on C-GQA in AUC. We include extended results including open-world setting and additional baselines in Appendix A.

| Method | MIT-States | UT-Zappos | C-GQA |
|---|---|---|---|
| ProtoProp | - | **34.7** | - |
| CGE | 6.5 | 33.5 | 4.2 |
| Co-CGE | 6.6 | 33.9 | 4.1 |
| CLIP | 11.0 | 5.0 | 1.4 |
| $\mathbb{CSP}$ | $\mathbf{19.4}_{0.1}$ | $33.0_{1.3}$ | $\mathbf{6.2}_{0.0}$ |

Table 4: Closed-world results comparing $\mathbb{CSP}$ with task-specific architectures. For $\mathbb{CSP}$, we report the average AUC on 5 random seeds with standard error.

**Generalization to Higher-Order Compositions**   To test the additional flexibility afforded by VLMs, we test if training $\mathbb{CSP}$ with attribute-object compositions can generalize to higher-order compositions such as attribute-attribute-object compositions.

We annotate a novel challenge dataset: AAO-MIT-States, a subset derived from the MIT-States dataset. In this dataset, we annotate the images in the test split of the MIT-States dataset with an additional attribute, to get an attribute-attribute-object pair as the class label. More details on the annotation are included in Appendix J.

| Method | Accuracy |
|---|---|
| CLIP | 62.7 |
| CoOp | $65.2_{0.3}$ |
| $\mathbb{CSP}$ | $\mathbf{72.6}_{0.4}$ |

Table 5: Unseen accuracy on 5 random seeds with std. error.

We compare CLIP, CoOp, and $\mathbb{CSP}$ to classify images with attribute-attribute-object classes. Since these class compositions are not present during training, we treat them as unseen classes and calculate the unseen accuracy. We take the best performing models for MIT-States and run inference on the challenge dataset.

Table 5 shows that $\mathbb{CSP}$ improves over CLIP by an average 9.9 percentage points on unseen accuracy and generalizes to attribute-attribute-object compositions without any modifications or training. The results demonstrate that $\mathbb{CSP}$ improves the compositionality of CLIP's vocabulary, even in ways that were not explicitly supervised.

**Generalization to Mixed Vocabulary**   To further test the additional flexibility afforded by VLMs, we also evaluate $\mathbb{CSP}$ on compositional zero-shot learning with a mixture of pretrained and fine-tuned vocabulary. This evaluation stems from the practical need to combine new unseen attributes with fine-tuned vocabulary. Evaluating in this setting will allow us to assess whether the benefits of $\mathbb{CSP}$ extend to classes including vocabulary not seen during fine-tuning. This setup goes beyond the above benchmarks, which include unseen *combinations* of attributes and objects, but all attributes and objects are seen during training. Now, we include completely unseen attributes.

We apply $\mathbb{CSP}$ on UT-Zappos with different fractions of attributes as seen attributes. We randomly select 25%, 50%, 75%, and 100% of the attributes and all the objects from the training set. Then, we remove from the seen classes the attribute-object pairs that include an unseen attribute. Finally, we train the on the remaining seen attribute-object pairs with five random seed values.

For each split of the seen and unseen attributes, we evaluate CSP by dividing the classes into three buckets: (1) unseen attribute + seen object pairs, (2) seen (i.e., fine-tuned) attribute + seen object pairs in unseen combinations, and (3) seen attribute + seen object pairs in seen combinations. In this evaluation, we refer to the classes in the first and second buckets as the unseen classes and those in the third bucket as the seen classes. This evaluation is more general than typical compositional zero-shot learning, which only evaluates on classes in the second and third buckets. Similar to our evaluation in Section 5, we add a scalar bias to the unseen classes and select the bias that maximizes the harmonic mean between accuracy on the unseen and seen classes in the validation set of UT-Zappos. We then report accuracy on unseen examples in each of the three buckets. To contextualize the performance of $\mathbb{CSP}$, we report the accuracy of CLIP on the unseen attribute + seen object pairs.

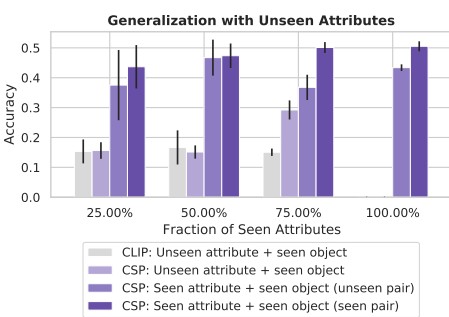

Figure 4: Results of $\mathbb{CSP}$ and CLIP with different fractions of pretrained and fine-tuned vocabulary. In each fraction, we report the average performance of CLIP and $\mathbb{CSP}$ on 5 random attribute splits.

Figure 4 shows that the performance on unseen attribute + seen object pairs improves with $\mathbb{CSP}$ and sufficient training pairs. Initially, the performance of CLIP and $\mathbb{CSP}$ are comparable but by providing more combinations of supervision for the objects $\mathbb{CSP}$ significantly outperforms CLIP on the unseen attribute + seen object evaluation bucket. These results demonstrate that the fine-tuned vocabulary of $\mathbb{CSP}$ can improve the compositional zero-shot performance of pretrained vocabulary.

**Additional Experiments** We summarize the additional experiments included in the Appendices B, C, D, and E.

We compare $\mathbb{CSP}$ with other parameter efficient methods (see Figure 5). In Appendix B, we experiment with CoCoOp, the conditional variant CoOp that incorporates visual information, on compositional zero-shot learning (Zhou et al., 2022). Our results show that $\mathbb{CSP}$ outperforms CoCoOp on all three datasets. In Appendix C, we compare CLIP adapter (Gao et al., 2021) and $\mathbb{CSP}$ on compositional zero-shot learning. We see that $\mathbb{CSP}$ still achieves the highest AUC across the datasets compared to CLIP adapters.

In Appendix D, to further test their flexibility, we see whether the learned vocabulary generalizes to classifying either attributes or objects alone. Our results show that $\mathbb{CSP}$ consistently improves attribute classification performance but can often reduce object classification accuracy.

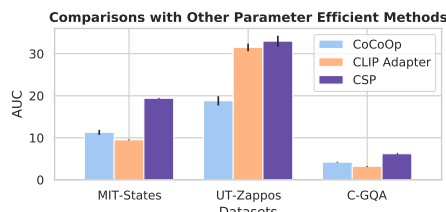

Figure 5: Closed-world results on MIT-States, UT-Zappos, and C-GQA comparing CoCoOp (Appendix B), CLIP adapters (Appendix C), and $\mathbb{CSP}$. We report the average AUC on 5 random seeds.

Finally, in Appendix E, we compare different ResNet and ViT backbones of CLIP. Our results show that $\mathbb{CSP}$ performance improves with larger backbones. In particular, we observe the highest gains with ViT backbones.

## 6 CONCLUSION

We present a new style of soft prompting, $\mathbb{CSP}$, for compositional zero-shot learning. We show that learning composable components of classes via soft prompting can improve downstream compositional zero-shot performance with a small number of parameters. We also demonstrate that the learned vocabulary generalizes in multiple, useful ways.

**Authors' Note.** The first two authors contributed equally. Co-first authors can prioritize their names when adding this paper's reference to their resumes.

## ACKNOWLEDGMENTS

We thank Andrew Delworth and Elise Carman for helping us annotate the AAO-MIT-States dataset. We appreciate the comments and advice from Cristina Menghini, Wasu Piriyakulkij, and Zheng-Xin Yong on our drafts. This material is based on research sponsored by Defense Advanced Research Projects Agency (DARPA) and Air Force Research Laboratory (AFRL) under agreement number FA8750-19-2-1006. The U.S. Government is authorized to reproduce and distribute reprints for Governmental purposes notwithstanding any copyright notation thereon. The views and conclusions contained herein are those of the authors and should not be interpreted as necessarily representing the official policies or endorsements, either expressed or implied, of Defense Advanced Research Projects Agency (DARPA) and Air Force Research Laboratory (AFRL) or the U.S. Government. We gratefully acknowledge support from Google and Cisco. Disclosure: Stephen Bach is an advisor to Snorkel AI, a company that provides software and services for weakly supervised machine learning.

## ETHICS STATEMENT

Zero-shot methods such as $\mathbb{CSP}$ should be thoroughly evaluated on in-distribution data before applying them to a downstream task. They should also be audited for biases, particuary if being used to make decisions that directly affect people. Models like CLIP are trained on Internet scale data contain biases that can cause harm to underrepresented groups, if not carefully evaluated. Further, any inference on unseen classes of data carries additional risk because of the domain shift between training and application. For more details, we refer the readers to Section 7 in Radford et al. (2021).

## REPRODUCIBILITY STATEMENT

The code for our experiments is available at https://github.com/BatsResearch/csp. We also show the pseudo-code for $\mathbb{CSP}$ in Appendix F. In Section 5, we provide all the relevant training details for the experiments: pretrained models, training and validation splits, GPUs, and python libraries. Finally, in Appendix H, we include the hyperparameters for all the datasets.

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

| | Method | MIT-States | | | | UT-Zappos | | | | C-GQA | | | |
|---|---|---|---|---|---|---|---|---|---|---|---|---|---|
| | | S | U | H | AUC | S | U | H | AUC | S | U | H | AUC |
| Closed | AoP(Nagarajan & Grauman, 2018) | 14.3 | 17.4 | 9.9 | 1.6 | 59.8 | 54.2 | 40.8 | 25.9 | 17.0 | 5.6 | 5.9 | 0.7 |
| | LE+ (Misra et al., 2017) | 15.0 | 20.1 | 10.7 | 2.0 | 53.0 | 61.9 | 41.0 | 25.7 | 18.1 | 5.6 | 6.1 | 0.8 |
| | TMN(Purushwalkam et al., 2019) | 20.2 | 20.1 | 13.0 | 2.9 | 58.7 | 60.0 | 45.0 | 29.3 | 23.1 | 6.5 | 7.5 | 1.1 |
| | SymNet(Li et al., 2020) | 24.2 | 25.2 | 16.1 | 3.0 | 49.8 | 57.4 | 40.4 | 23.4 | 26.8 | 10.3 | 11.0 | 2.1 |
| | CompCos (Mancini et al., 2021a) | 25.3 | 24.6 | 16.4 | 4.5 | 59.8 | 62.5 | 43.1 | 28.1 | 28.1 | 11.2 | 12.4 | 2.6 |
| | ProtoProp (Ruis et al., 2021) | - | - | - | - | 62.1 | 65.5 | 50.2 | **34.7** | - | - | - | - |
| | CGE (Naeem et al., 2021) | 32.8 | 28.0 | 21.4 | 6.5 | **64.5** | **71.5** | **60.5** | 33.5 | **33.5** | 15.5 | 16.0 | 4.2 |
| | Co-CGE (Mancini et al., 2021b) | 32.1 | 28.3 | 20.0 | 6.6 | 62.3 | 66.3 | 48.1 | 33.9 | 33.3 | 14.9 | 15.5 | 4.1 |
| | CLIP (Radford et al., 2021) | 30.2 | 46.0 | 26.1 | 11.0 | 15.8 | 49.1 | 15.6 | 5.0 | 7.5 | 25.0 | 8.6 | 1.4 |
| | CoOp (Zhou et al., 2021) | $34.4_{0.1}$ | $47.6_{0.1}$ | $29.8_{0.1}$ | $13.5_{0.0}$ | $52.1_{0.5}$ | $49.3_{1.8}$ | $34.6_{1.7}$ | $18.8_{1.4}$ | $20.5_{0.2}$ | $\mathbf{26.8}_{0.3}$ | $17.1_{0.2}$ | $4.4_{0.1}$ |
| | $\mathbb{CSP}$ (Ours) | $\mathbf{46.6}_{0.1}$ | $\mathbf{49.9}_{0.1}$ | $\mathbf{36.3}_{0.1}$ | $\mathbf{19.4}_{0.1}$ | $64.2_{0.7}$ | $66.2_{1.2}$ | $46.6_{1.2}$ | $33.0_{1.3}$ | $28.8_{0.1}$ | $\mathbf{26.8}_{0.1}$ | $\mathbf{20.5}_{0.1}$ | $\mathbf{6.2}_{0.0}$ |
| Open | AoP(Nagarajan & Grauman, 2018) | 16.6 | 5.7 | 4.7 | 0.7 | 50.9 | 34.2 | 29.4 | 13.7 | - | - | - | - |
| | LE+ (Misra et al., 2017) | 14.2 | 2.5 | 2.7 | 0.3 | 60.4 | 36.5 | 30.5 | 16.3 | 19.2 | 0.7 | 1.0 | 0.08 |
| | TMN(Purushwalkam et al., 2019) | 12.6 | 0.9 | 1.2 | 0.1 | 55.9 | 18.1 | 21.7 | 8.4 | - | - | - | - |
| | SymNet(Li et al., 2020) | 21.4 | 7.0 | 5.8 | 0.8 | 53.3 | 44.6 | 34.5 | 18.5 | 26.7 | 2.2 | 3.3 | 0.43 |
| | CompCos (Mancini et al., 2021a) | 21.4 | 7.0 | 5.8 | 0.8 | 53.3 | 44.6 | 34.5 | 18.5 | 26.7 | 2.2 | 3.3 | 0.43 |
| | CGE (Naeem et al., 2021) | 32.4 | 5.1 | 6.0 | 1.0 | 61.7 | **47.7** | 39.0 | 23.1 | **32.7** | 1.8 | 2.9 | 0.47 |
| | Co-CGE$^{CW}$ (Mancini et al., 2021b) | 31.1 | 5.8 | 6.4 | 1.1 | 62.0 | 44.3 | 40.3 | 23.1 | 32.1 | 2.0 | 3.4 | 0.53 |
| | Co-CGE$^{open}$ (Mancini et al., 2021b) | 30.3 | 11.2 | 10.7 | 2.3 | 61.2 | 45.8 | **40.8** | **23.3** | 32.1 | 3.0 | 4.8 | 0.78 |
| | CLIP (Radford et al., 2021) | 30.1 | 14.3 | 12.8 | 3.0 | 15.7 | 20.6 | 11.2 | 2.2 | 7.5 | 4.6 | 4.0 | 0.27 |
| | CoOp (Zhou et al., 2021) | $36.8_{0.1}$ | $\mathbf{16.5}_{0.1}$ | $16.1_{0.1}$ | $4.7_{0.0}$ | $61.8_{0.5}$ | $39.3_{1.3}$ | $35.6_{0.7}$ | $19.5_{0.6}$ | $20.9_{0.3}$ | $4.5_{0.2}$ | $5.7_{0.2}$ | $0.73_{0.0}$ |
| | CoOp (Zhou et al., 2021) | $34.6_{0.1}$ | $9.3_{0.0}$ | $12.3_{0.1}$ | $2.8_{0.0}$ | $52.1_{0.5}$ | $31.5_{2.9}$ | $28.9_{2.3}$ | $13.2_{1.6}$ | $21.0_{0.2}$ | $4.6_{0.1}$ | $5.5_{0.1}$ | $0.70_{0.0}$ |
| | $\mathbb{CSP}$ (Ours) | $\mathbf{46.3}_{0.3}$ | $15.7_{0.1}$ | $\mathbf{17.4}_{0.1}$ | $\mathbf{5.7}_{0.0}$ | $\mathbf{64.1}_{0.7}$ | $44.1_{0.3}$ | $38.9_{0.5}$ | $22.7_{0.4}$ | $28.7_{0.2}$ | $\mathbf{5.2}_{0.1}$ | $\mathbf{6.9}_{0.1}$ | $\mathbf{1.20}_{0.0}$ |

Table 6: Closed-world (Closed) results on MIT-States, UT-Zappos, and C-GQA. For CoOp and $\mathbb{CSP}$, we report the average performance of the models on 5 random seeds with standard error. The results for AoP, LE+, TMN, SymNet, CompCos, CGE, and Co-CGE are obtained from Mancini et al. (2021b) and ProtoProp from Ruis et al. (2021). For extended results, see Appendix A.

# A    COMPARISON OF EXISTING TASK-SPECIFIC ARCHITECTURES

In this section, we compare $\mathbb{CSP}$ to existing task-specific compositional zero-shot learning methods.

**Baselines**    We consider the following compositional zero-shot learning methods in closed-world and open-world setting: AoP (Nagarajan & Grauman, 2018), LE+ (Misra et al., 2017), TMN (Purushwalkam et al., 2019), SymNet (Li et al., 2020), CompCos (Mancini et al., 2021a), CGE (Naeem et al., 2021), and Co-CGE (Mancini et al., 2021b). We also consider ProtoProp (Ruis et al., 2021) in the closed-world setting.

**Results**    Our results in Table 6 show that $\mathbb{CSP}$ outperform existing compositional zero-shot learning method on MIT-States and C-GQA while being competitive on UT-Zappos in AUC. In the closed-world setting, $\mathbb{CSP}$ improves over existing compositional zero-shot learning methods by 12.8 points on MIT-states and 2.0 points on C-GQA in AUC. In the open-world setting, we improve over the existing compositional zero-shot learning methods on MIT-States by 3.4 points and C-GQA by 0.42 points in AUC.

# B    CONDITIONAL VARIANT OF COOP AND CSP

We experiment with CoCoOp, the conditional variant of CoOp, on compositional zero-shot learning. CoCoOp showed improved performance over CoOp in few-shot object classification by using visual information to condition the prompts. We investigate if additional visual information in prompts can help in compositional zero-shot learning.

CoCoOp uses a lightweight network to generate an image-specific bias vector and adds them to the learnable vocabulary in the prompt. For fair comparison, we also extend $\mathbb{CSP}$ to CoCSP to incorporate visual information into the prompts.

**Setup**    We train CoCoOp and CoCSP with the same hyperparmeters in Appendix H. Additionally, the lightweight network in our experiments is a two-layer multilayer perceptron with a ReLU activation between the two layers. The input and the output dimensions of the network are $d$ and the hidden dimension is $d/16$ where $d$ is the dimension of the vocabulary. The generated image-specific bias vector is added to the context vocabulary in CoCoOp and attribute-object vocabulary in CoCSP.

| Method | *MIT-States* | *UT-Zappos* | *C-GQA* |
|--------|--------------|-------------|---------|
| CLIP | 11.0 | 5.0 | 1.4 |
| CoOp | 13.5 $_{0.0}$ | 18.8 $_{1.4}$ | 4.4 $_{0.1}$ |
| $\mathbb{CSP}$ | **19.4** $_{0.1}$ | **33.0** $_{1.3}$ | **6.2** $_{0.0}$ |
| CoCoOp | 11.3 $_{0.6}$ | 18.8 $_{1.1}$ | 4.2 $_{0.1}$ |
| CoCSP | 17.6 $_{0.1}$ | 32.7 $_{0.3}$ | 5.7 $_{0.0}$ |

Table 7: Closed-world results on MIT-States, UT-Zappos, and C-GQA. For $\mathbb{CSP}$ , CoOp, CoCoOp, and CoCSP, we report the average AUC on 5 random seeds with standard error. * denotes a bug in the soft prompt for CoCoOp which we will fix in the final version. In CoCoOp for C-GQA, we average the pre-trained vocabulary for attributes and objects with multiple tokens instead of using them as is in the prompt.

**Results**    Table 7 shows CoCSP improves upon CoCoOp across the three datasets. We also observe no benefits over non-conditional methods such as $\mathbb{CSP}$ and CoOp in AUC. We suspect that additional parameters for compositional generalization from seen pairs of concepts to unseen pairs.

## C   CLIP ADAPTERS WITH COMPOSITIONAL SOFT PROMPTS

We extend CLIP adapter (Gao et al., 2021) to compositional zero-shot learning. Adapters are an alternate method of parameter-efficient learning for large-scale models that has shown improved performance on several downstream tasks (Houlsby et al., 2019).

CLIP adapters are a variant of the adapters for few-shot object classification (Houlsby et al., 2019). Instead of adding small feedforward networks to all the layers, they learn a multilayer perceptron in the image encoder and the text encoder. They transform the image representation from the encoder with the multilayer perceptron and include a residual connection from the image encoder and the same for the text representation. While the CLIP adapters can be added to the text encoder, the authors show that visual adapters perform better than textual adapters. For this reason, we compare with visual adapters in the experiment.

**Setup**    We add a CLIP adapter, a two-layer multilayer perceptron with ReLU activation, to the image encoder. We use the same prompt prompt as $\mathbb{CSP}$ and train only the CLIP adapter. For fair comparison, we also train a model with $\mathbb{CSP}$ and CLIP adapters where we jointly train the prompts and adapter on the task. We train with the same hyperparamters settings for CLIP adapters and CLIP adapters + $\mathbb{CSP}$ in Appendix H.

| Method | *MIT-States* | *UT-Zappos* | *C-GQA* |
|--------|--------------|-------------|---------|
| CLIP | 11.0 | 5.0 | 1.4 |
| CLIP adapter | 9.5 $_{0.1}$ | 31.5 $_{0.9}$ | 3.2 $_{0.1}$ |
| CLIP adapter + $\mathbb{CSP}$ | 8.3 $_{0.2}$ | 32.5 $_{1.0}$ | 2.7 $_{0.0}$ |
| $\mathbb{CSP}$ | **19.4** $_{0.1}$ | **33.0** $_{1.3}$ | **6.2** $_{0.0}$ |

Table 8: Closed-world results on MIT-States, UT-Zappos, and C-GQA. For CLIP adapter, CLIP adapter + $\mathbb{CSP}$ , and $\mathbb{CSP}$ , we report the average AUC on 5 random seeds with standard error.

**Results**    Table 8 shows that CLIP adapters can improve performance over CLIP but $\mathbb{CSP}$ performs better on its own. We also observe that combining CLIP adapters with $\mathbb{CSP}$ can often hurt performance in AUC.

## D   DECOMPOSITION OF ATTRIBUTES AND OBJECTS

To further test the flexibility of $\mathbb{CSP}$, we investigate how the learned vocabulary performs on attribute classification and object classification separately. We create prompts of the form `a photo of [attribute] object` and `a photo of [object]` classify images according to either attribute or

|   | Method | MIT-States | | UT-Zappos | | C-GQA | |
|---|--------|------------|---|-----------|---|-------|---|
|   |        | A | O | A | O | A | O |
| S | CLIP | 16.6 | 44.3 | 8.8 | 58.5 | 3.6 | 25.4 |
|   | CoOp | $18.7_{0.2}$ | $\mathbf{47.6}_{0.1}$ | $25.8_{3.4}$ | $57.8_{3.1}$ | $7.0_{0.6}$ | $\mathbf{26.3}_{0.4}$ |
|   | $\mathbb{CSP}$ | $\mathbf{24.5}_{0.2}$ | $40.2_{0.2}$ | $\mathbf{66.4}_{0.5}$ | $\mathbf{63.8}_{1.2}$ | $\mathbf{12.5}_{0.1}$ | $24.2_{0.1}$ |
| U | CLIP | 19.4 | 46.8 | 11.1 | $\mathbf{65.6}$ | 4.1 | 25.4 |
|   | CoOp | $22.0_{0.2}$ | $\mathbf{49.6}_{0.1}$ | $21.5_{6.5}$ | $37.1_{2.9}$ | $5.9_{0.5}$ | $\mathbf{25.9}_{0.4}$ |
|   | $\mathbb{CSP}$ | $\mathbf{23.3}_{0.1}$ | $36.3_{0.1}$ | $\mathbf{49.2}_{1.8}$ | $46.6_{3.6}$ | $\mathbf{6.0}_{0.1}$ | $23.8_{0.1}$ |

Table 9: Results for decomposition of learned attribute and object vocabulary. We report average top-1 accuracy for attribute (A) and object (O) classification on 5 random seeds with standard error. Evaluation is done on seen (S) classes and unseen (U) classes of each dataset.

object. We measure accuracy on two sets of evaluation data. The first is the seen classes of each dataset, meaning that the classification is performed on new examples of attributes and objects that previously have been seen in the same combination. The second is the unseen classes of the original datasets, so this can be thought of as a kind of domain shift problem, evaluating how well the learned primitive concepts can be reused in isolation on novel attribute-object combinations.

Table 9 shows that $\mathbb{CSP}$ consistently improves attribute classification performance, while often reducing object classification accuracy. Sometimes CoOp improves over CLIP's object classification accuracy as well. This results indicates that $\mathbb{CSP}$ learns useful standalone attribute representations that are also composable with objects, but that the resulting object representations might not be as good on their own.

## E    MODEL ABLATION

Table 10 shows that $\mathbb{CSP}$ with a performance improves with larger backbones. We note that $\mathbb{CSP}$ generally improves performance over CLIP. In particular, we see the gains are highest with ViT backbones.

| Method | Backbone | MIT-States | | | | UT-Zappos | | | | C-GQA | | | |
|--------|----------|---|---|---|-----|---|---|---|-----|---|---|---|-----|
|        |          | S | U | H | AUC | S | U | H | AUC | S | U | H | AUC |
| CLIP | ResNet-50 | 21.1 | 34.4 | 18.4 | 5.6 | 6.4 | 43.6 | 6.4 | 1.4 | 6.1 | 17.1 | 6.1 | 0.7 |
| CLIP | ResNet-101 | 25.2 | 37.4 | 21.7 | 7.5 | 11.2 | 35.2 | 11.9 | 2.8 | 7.3 | 19.7 | 7.6 | 1.1 |
| CLIP | ViT B/32 | 25.1 | 39.1 | 21.4 | 7.5 | 9.6 | 42.4 | 10.0 | 2.4 | 7.3 | 22.1 | 7.4 | 1.2 |
| CLIP | ViT L/14 | 30.2 | 46.0 | 26.1 | 11.0 | 15.8 | 49.1 | 15.6 | 5.0 | 7.5 | 25.0 | 8.6 | 1.4 |
| $\mathbb{CSP}$ | ResNet-50 | $35.0_{0.1}$ | $30.3_{0.1}$ | $23.0_{0.1}$ | $8.3_{0.1}$ | $21.8_{1.6}$ | $11.3_{1.7}$ | $10.2_{1.2}$ | $1.8_{0.3}$ | $17.9_{0.3}$ | $14.7_{0.2}$ | $10.2_{0.2}$ | $1.8_{0.0}$ |
| $\mathbb{CSP}$ | ResNet-101 | $38.9_{0.1}$ | $32.1_{0.2}$ | $25.2_{0.1}$ | $9.9_{0.1}$ | $42.2_{1.2}$ | $9.1_{1.3}$ | $10.0_{1.1}$ | $2.6_{0.5}$ | $17.7_{0.1}$ | $17.0_{0.2}$ | $12.0_{0.1}$ | $2.3_{0.0}$ |
| $\mathbb{CSP}$ | ViT B/32 | $36.4_{0.4}$ | $42.5_{0.2}$ | $28.6_{0.1}$ | $12.4_{0.1}$ | $57.1_{0.4}$ | $57.3_{0.6}$ | $39.3_{0.6}$ | $24.2_{0.4}$ | $\mathbf{30.1}_{0.1}$ | $23.4_{0.2}$ | $19.4_{0.3}$ | $5.7_{0.1}$ |
| $\mathbb{CSP}$ | ViT L/14 | $\mathbf{46.6}_{0.1}$ | $\mathbf{49.9}_{0.1}$ | $\mathbf{36.3}_{0.1}$ | $\mathbf{19.4}_{0.1}$ | $\mathbf{64.2}_{0.7}$ | $\mathbf{66.2}_{1.2}$ | $\mathbf{46.6}_{1.2}$ | $\mathbf{33.0}_{1.3}$ | $28.8_{0.1}$ | $\mathbf{26.8}_{0.1}$ | $\mathbf{20.5}_{0.1}$ | $\mathbf{6.2}_{0.0}$ |

Table 10: Closed-world ablation results with respect to different backbone architectures of CLIP. We report the average performance of the model on 5 random seeds with standard error.

## F    PSEUDOCODE

Figure 6 shows the Torch-like pseudocode for inference with $\mathbb{CSP}$. The function accepts the minibatch of images, test pairs, and the clip model with the fine-tuned embeddings and returns the cosine similarities between the image representation and the text representation scaled by a constant scalar.

## G    FEASIBILITY CALIBRATION FOR OPEN-WORLD SETTING

Feasibility calibration aims to filter out infeasible compositions that might be present in the open-world setting. To filter out infeasible compositions, we follow the post-training calibration from Mancini et al. (2021b). They conjecture that similar objects share similar attributes while dissimilar objects are unlikely to share attributes. For example, `cat` and `dog` can share the attribute `old` but `cat` and `cliff` do not share the attribute `eroded`.

```python
def inference(batch_images: nn.Tensor,
              test_pairs: List[List, List],
              model: nn.Module):
    """
    Function to run inference with the fine-tuned embeddings.
    Args:
        batch_images (torch.Tensor): minibatch of images [n, h, w, c]
        test_pairs (tuple): attribute-object pairs in the test
            split [m, 2]
        model (nn.Module): model with the fine-tuned embeddings
    Returns:
        torch.Tensor: cosine similarties of the minibatch images
            and attribute-object pairs [n, m]
    """
    prompt_template = "a photo of x x"
    tokenized_prompt = tokenize(prompt_template)
    tokenized_prompt = tokenized_prompt.repeat(len(test_pairs))
    token_tensor = model.token_embedding(tokenized_prompt)

    # fine-tuned embeddings
    attr, obj = zip(*test_pairs)
    attr_emb = model.soft_embedding(attr)
    obj_emb = model.soft_embedding(obj)

    # replace the "x x" in prompt template with fine-tuned embeddings
    token_tensor = replace_emb(token_tensor, attr_emb, obj_emb)

    # l2-normalized
    text_rep = model.text_encoder(token_tensor)
    image_rep = model.image_encoder(batch_images)

    logits = (image_rep @ text_rep) * model.logit_scale.exp()

    return logits
```

Figure 6: Torch-like pseudocode for inference with $\mathbb{CSP}$.

We calculate the feasibility compositions for the composition $(a, o)$ by computing the relationships between the objects and the attributes. First, we find the similarities between the objects:

$$\rho_o(a, o) = \max_{\hat{o} \in \mathbb{O}_{seen}} \frac{\phi(o) \cdot \phi(\hat{o})}{||\phi(o)|| \, ||\phi(\hat{o})||}$$

where $\rho_o(.)$ is the similarity between the object $o$ with other objects $\hat{o}$ and $\phi(.)$ is an embedding function that maps attributes to pretrained embedding. We compute similarities between the attributes in the same way.

Next, we combine the two similarities with a pooling function. In our case, we use mean pooling $\mu$:

$$\rho(a, o) = \mu(\rho_o(a, o), \rho_a(a, o))$$

where $\rho(a, o)$ is the feasbility score for the composition $(a, o)$. Finally, we filter out infeasible compositions by considering compositions above a threshold $T$ calibrated on the validation set to get our final prediction:

$$\hat{y} = \underset{y \in \mathbb{Y}_{test}, \, \rho(a,o) > T}{\arg\max} p_{\boldsymbol{\theta}}(y \mid x)$$

Following prior work, we compute feasibility calibration using GloVe embeddings (Pennington et al., 2014), and filter out the infeasible attribute-object compositions based on the performance on the validation split.

## H  HYPERPARAMETERS

In our work, we find the best hyperparameters for training $\mathbb{CSP}$ via a grid search. We train the ViT-B/32 model for 50 epochs and use the same best performing hyperparameters to train all our models including ViT L/14. We run a grid search with the following hyperparameters: (1) learning rate: $\{5e - 03, 5e - 04, 5e - 05\}$, (2) batch size: $\{128, 256\}$, (3) attribute dropout: $\{0.0, 0.1, 0.2, 0.3\}$, and (4) weight decay: $\{1e - 05, 5e - 05\}$. We choose the hyperparameters for a dataset based best unseen accuracy on the validation split. We reduce the number of epochs to 20 with ViT L/14 as we found our models tend to converge earlier.

| Hyperparameter | *MIT-States* | *UT-Zappos* | *C-GQA* |
|---|---|---|---|
| Learning rate | $5e-05$ | $5e-04$ | $5e-05$ |
| Batch size | 128 | 128 | 128 |
| Attribute dropout | 0.3 | 0.2 | 0.3 |
| Weight decay | $1e-05$ | $1e-05$ | $5e-05$ |

Table 11: Hyperparameters for MIT-States, UT-Zappos, and C-GQA.

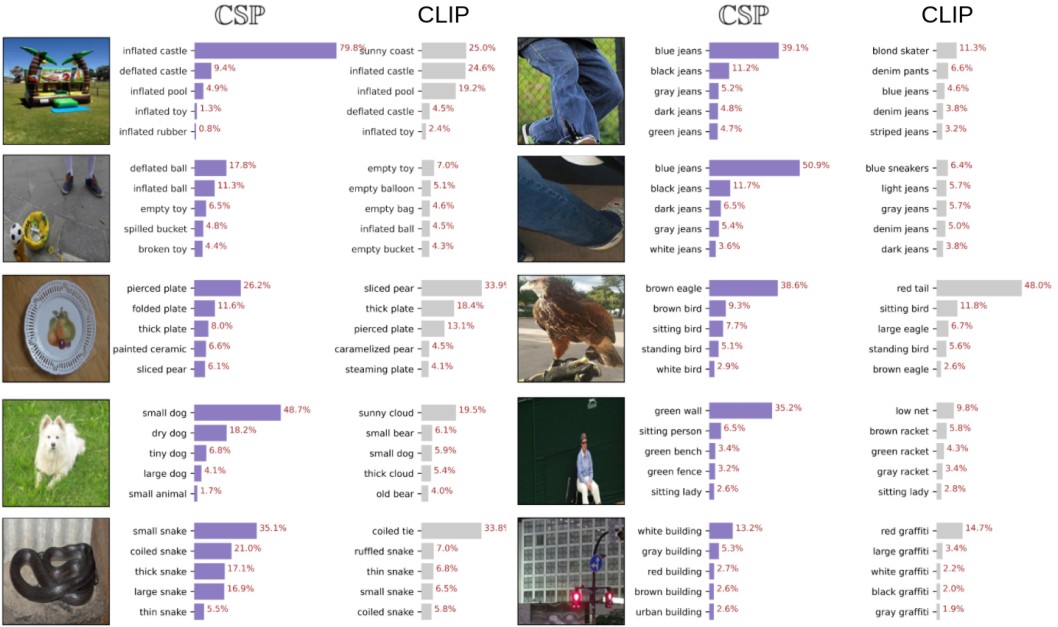

Figure 7: Additional qualitative comparison for image to text retrieval between $\mathbb{CSP}$ and CLIP on CGQA. Selected samples with concepts correctly identified and top-5 retrieval results by $\mathbb{CSP}$ are shown.

Table 11 shows the hyperparameters used to train $\mathbb{CSP}$ on all the datasets. The experiments with CoOp and CLIP adapters do not use dropout as the original architecture did not use dropout in the text or the image encoder. For the rest of the architecture, we use the same hyperparameters for CoOp and CLIP adapters as $\mathbb{CSP}$.

## I  ADDITIONAL QUALITATIVE EXAMPLES

In this section, we include additional examples from the compositional zero-shot image to text retrieval tasks. Selected samples from CGQA in Figure 7 show that fine-tuning the vocabulary enables $\mathbb{CSP}$ to better identify composed concepts compared to CLIP. In particular, we observe that $\mathbb{CSP}$ ranks relevant attributes in the attribute-object composition better than CLIP.

## J  DATASET CREATION

We create AAO-MIT-States from the MIT-States dataset (Isola et al., 2015). Below we include details on the annotation interface, annotators, and aggregation process for the annotations.

We annotate an additional attribute for the images paired with unseen classes in the test split of MIT-States. The interface has three main components: (1) general instructions, (2) image with caption, and (3) list of populated attributes. The general instructions provide the annotators with a detailed description of the annotation task. To the right of the instructions is a randomly sampled

image from the test split of MIT-States. Additionally, we include a caption describing the image. For example, suppose we select an image of a wet cat, we ask the users: "which attribute best describes the cat in the image presented?". Since we have a large number of attributes in the MIT-States dataset, we need a way to reduce the list of attributes the user observes while annotating a single image. We use CLIP to predict the attributes except for the original attribute in the image and choose the top-5 attributes as annotation candidates. We also include an option to select none of the above.

Figure 8: Example annotation interface.

The dataset was annotated by two of the authors and two undergraduate research assistants. We randomly sampled a total of 1200 images from test-split and asked the annotators to annotate from the interface. We received annotations for 1089 instances where each image received exactly three annotations. We aggregate examples for our dataset where all the three annotators agreed on the same attribute other than "None of the above". The total number of examples in the final annotated dataset is 193. We have open-sourced the dataset in our code.

