# OpenReview forum: "Learning to Compose Soft Prompts for Compositional Zero-Shot Learning"
_ICLR.cc/2023/Conference — ICLR 2023 poster_

### Official Review · Reviewer_m1LH · 2022-10-20

**Confidence:** 4
**Correctness:** 4
**Technical Novelty And Significance:** 3
**Empirical Novelty And Significance:** 3
**Recommendation:** 8

**Clarity, Quality, Novelty And Reproducibility:**

**Clarity**: Overall, the paper is well-written and easy to follow.

**Quality**: The paper provides comprehensive experiments, which are helpful for readers to understand what benefits the simple prompt learning approach could bring to the area of compositional zero-shot learning. The method is well-motivated and has a simple design, which could be inspiring to the community.

**Novelty**: The research is generally novel because previous work often uses task-specific architectures while this work shows that by applying the simple prompt learning approach to large vision-language models, we can obtain non-trivial improvements. The method and findings could be of interest to both the community of compositional zero-shot learning and those who are working on large-scale models and prompt learning.

**Reproducibility**: The paper seems to contain sufficient details for readers to re-implement the method and reproduce the results.

**Strength And Weaknesses:**

**Strengths**

The emergence of vision foundation models, particularly vision-language models like CLIP, has offered many opportunities to the computer vision community. This work provides timely insights on how prompt learning, a simple technique that tunes a few parameters in a model's input space, can be used to tackle compositional zero-shot learning. Though soft prompt learning has been applied to CLIP before, the results shown in this work on the compositional zero-shot learning setting are novel and useful to the community.

It is also worth mentioning that the field of compositional zero-shot learning has been previously dominated by task-specific methods. This work shows that a simple technique based on prompt learning and a large-scale vision-language model could provide significant improvements to the problem. Given the importance of large-scale models, which have become a trend in the community, this work could serve as a strong baseline to build upon for future work.

**Weaknesses**

Below some minor comments are listed for the authors to address.

1. From the technical point of view, CoOp fine-tunes all context tokens while this work fine-tunes only the attribute token and the object class token. It's good to see that such a seemingly trivial twist can lead to huge improvement. Currently, the paper doesn't provide in-depth analysis in this regard. Would it be possible for the authors to elabroate on this design and explain (perhaps with some evidence) why such a small change could make a big difference?

2. The paper only studies this template, "a photo of xxx". Have the authors tried other variants and if so, what are observed in the results?

**Summary Of The Paper:**

The paper addresses the compositional zero-shot learning problem using CLIP, a powerful vision-language model that was pretrained on broad data to measure similarity between images and texts. Instead of fine-tuning the entire model, the paper proposes to fine-tune only the attribute and object tokens in a prompt template in the text input space of CLIP. Extensive experiments on three benchmarks, i.e., MIT-states, UT-Zappos and C-GQA, show that the proposed method, despite having a simple design, outperforms several other relevant baselines.

**Summary Of The Review:**

The paper provides novel insights on how to apply prompt learning to large vision-language models to solve compositional zero-shot learning. Overall, the paper is well-written, the method is simple and technically sound, and the results are encouraging and useful. So I recommend to accept the paper.

== Post-rebuttal update ==

The authors have done a good job in addressing the two weakness points. My view about the novelty, results and significance remains the same: the paper should be accepted.

I have also read other reviewers' comments and noticed that some reviewers have concerns about the technical novelty and the resources needed to further tune CLIP. I would like to speak for the authors here.
- The paper should be evaluated in the context of compositional zero-shot learning. The novelty is then clear because existing methods are mostly based on task-specific models while this paper demonstrates the effectiveness of such a simple prompt-driven paradigm, which is encouraging and could spur a paradigm shift in compositional zero-shot learning.
- CLIP was indeed trained with massive data but this doesn't mean it must work in any scenario without some adaptation—which is the incomplete nature of foundation models. It would make more sense if we take a look at the counterpart in NLP like GPT or BERT. To adapt large language models, one also needs to apply some adaptation methods, such as learning specialized modules or using prompt learning.
- It might seem straightforward to apply task-specific learning (e.g., prompt learning) to adapting CLIP to solve compositional zero-shot learning, but such a "straightforward" hypothesis has never been scientifically validated—which is done in this work.

---

> ### Author Response · Authors · 2022-11-13
> **Response to Reviewer m1LH**
>
> Thank you for your thoughtful comments on our manuscript. Below we address your comments and questions about our work.
>
> > From the technical point of view, CoOp fine-tunes all context tokens while this work fine-tunes only the attribute token and the object class token. It's good to see that such a seemingly trivial twist can lead to huge improvement. Currently, the paper doesn't provide in-depth analysis in this regard. Would it be possible for the authors to elabroate on this design and explain (perhaps with some evidence) why such a small change could make a big difference?”
>
> You bring up a great point. The key difference between CoOP and CSP is the compositional training of soft prompts. CoOp trains a single prompt on the downstream task whereas CSP trains multiple soft prompt compositions on the downstream task. We suspect that explicitly enforcing compositional semantics during fine-tuning is the key. We also see that our results in Figure 4 show that increasing the number of compositions seen at fine-tuning time improves generalization.
>
> > The paper only studies this template, "a photo of xxx". Have the authors tried other variants and if so, what are observed in the results?”
>
> Thank you for pointing this out. We have conducted additional experiments with five different prompt templates. Our results below confirm that CLIP might only be marginally sensitive to prompts in this setting.
>
> | (AUC)            | MIT-States | UT-Zappos | CGQA |
> |------------------|------------|-----------|------|
> | **A photo of**   | 11.0       | 5.0       | 1.4  |
> | a composition of | 11.1       | 4.4       | 1.3  |
> | a picture of     | 11.3       | 4.9       | 1.2  |
> | a photo of many  | 10.8       | 4.3       | 1.4  |
> | an image of      | 11.2       | 3.9       | 1.4  |
> | A                | 11.0       | 3.2       | 1.4  |

---

> > ### Comment · Reviewer_m1LH · 2022-11-19
> > **Good work**
> >
> > Thanks for the rebuttal. I have updated my review and keep the rating unchanged.

---

### Official Review · Reviewer_43mh · 2022-10-22

**Confidence:** 4
**Clarity, Quality, Novelty And Reproducibility:** I wrote above my concerns regarding t…
**Correctness:** 3
**Technical Novelty And Significance:** 3
**Empirical Novelty And Significance:** 3
**Recommendation:** 6

**Details Of Ethics Concerns:**

No.

**Strength And Weaknesses:**

**Strengths:**

1) Overall, this paper is well written, and the technical details are easy to follow.


2) The main idea of introducing better prompting techniques for zero-shot compositionality is appealing.


3) The main contribution of this paper is the compositional soft prompting technique, which treats the attributes and objects that are composed to define classes as learnable tokens of vocabulary in a prompt as “A photo of [attribute] [object]”. Although I consider this a minor novelty in general, it still meets the bar.

4) The results of the experiment strongly support the proposed approach, including the strong ablations that were performed.


**Weaknesses:**

**Experiments.** Recently, prompting has become a hot topic. Why not add more recent prompting techniques such as VPT [1] and others? Essentially, the proposed technique is a standard CLIP training with additional attribute and object prompting. Therefore, I believe it would be useful to have a more comprehensive comparison of prompting baselines.

[1] Visual Prompt Tuning, ECCV 2022

**Technical Novelty.** As I said above, the main contribution of the paper is the treatment of the attributes and objects that are composed to define classes as learnable tokens of vocabulary in a prompt as “A photo of [attribute] [object]”. I find it difficult to consider the proposed approach (prompting of attributes and objects together) to be a significant contribution, but I acknowledge that it is the first paper to address this important issue.


**Relation to Prior Work.** There are several works in that area based on compositionality for videos or scene graphs, which the author could also add and discuss. This topic is very important, and the related work paragraph has a limited scope, which is fine, but here are some additional references the authors may wish to consider:

[1] Bar et al. Compositional Video Synthesis with Action Graphs, ICML 2021.

[2] Generating videos of zero-shot compositions of actions and objects. ECCV 2020.

[3] Compositional video prediction. ICCV 2019.

[4] Learning Canonical Representations for Scene Graph to Image Generation. ECCV 2020.




**Summary Of The Paper:**

This paper presents compositional soft prompting (CSP), a parameter-efficient learning technique to improve the zero-shot compositionality of a large-scale pretrained vision-language model. In addition, the paper demonstrates that the learned vocabulary generalizes across multiple datasets. Finally, the proposed method substantially improves the results on multiple datasets.

**Summary Of The Review:**

My main concern is that this paper presented a minor approach to tackles the important problem of the compositionality of VL models. I am open to the authors' feedback and other reviewers' opinions.

---

> ### Author Response · Authors · 2022-11-13
> **Response to Reviewer 43mh**
>
> Thank you for your thoughtful comments on our manuscript. Below we address your comments and questions about our work.
>
> ​​> Experiments. Recently, prompting has become a hot topic. Why not add more recent prompting techniques such as VPT [1] and others? Essentially, the proposed technique is a standard CLIP training with additional attribute and object prompting. Therefore, I believe it would be useful to have a more comprehensive comparison of prompting baselines.
> >
> > [1] Visual Prompt Tuning, ECCV 2022
>
> Thank you for your suggestion. We focused on prompting techniques that tune the text encoder because new, unseen classes at test time are defined using language. Without tuning the text encoder, the representation of the classes is determined by the pre-training, no matter how the image encoder is tuned.
>
> As suggested, we also perform an experiment with visual prompt tuning. We use shallow prompting with a prompt length of 4 as it is similar to CoOp. We follow the same initialization scheme and pass the learnable visual tokens to the transformer layers. Our results, below, show that visual prompt tuning does not generalize well to the compositional zero-shot learning task. In fact, both CoOp and CSP perform significantly better than visual prompt tuning.
>
> |                            | MIT-States | UT-Zappos | C-GQA |
> |----------------------------|------------|-----------|-------|
> | CLIP                       | 11.0       | 5.0       | 1.4   |
> | Visual prompt tuning (VPT) | 11.1       | 11.9      | 1.4   |
> | CoOp                       | 13.5       | 18.8      | 4.4   |
> | CSP                        | 19.4       | 33.0      | 6.2   |
>
>
> > Technical Novelty. As I said above, the main contribution of the paper is the treatment of the attributes and objects that are composed to define classes as learnable tokens of vocabulary in a prompt as “A photo of [attribute] [object]”. I find it difficult to consider the proposed approach (prompting of attributes and objects together) to be a significant contribution, but I acknowledge that it is the first paper to address this important issue.
>
> Thank you, we agree that CSP addresses an important issue. Even though the proposed approach is simple to implement and understand, we believe that demonstrating that it is actually effective is a significant contribution. As you say, no one has studied it before.
>
> In terms of broader significance, Reviewer m1LH says that our findings are of significant interest to researchers studying compositionality. We also believe that there are many domains where learning specialized, composable vocabularies on top of pre-trained models will be valuable.
>
> > Relation to Prior Work. There are several works in that area based on compositionality for videos or scene graphs, which the author could also add and discuss. This topic is very important, and the related work paragraph has a limited scope, which is fine, but here are some additional references the authors may wish to consider:
> >
> > [1] Bar et al. Compositional Video Synthesis with Action Graphs, ICML 2021.
> >
> > [2] Generating videos of zero-shot compositions of actions and objects. ECCV 2020.
> >
> > [3] Compositional video prediction. ICCV 2019.
> >
> > [4] Learning Canonical Representations for Scene Graph to Image Generation. ECCV 2020.
>
> Thank you for suggesting these papers. We have updated our manuscript citing these works.

---

> > ### Comment · Reviewer_43mh · 2022-11-19
> > **Response**
> >
> > After reading the authors' feedback and other reviewers' opinions, I would like to thank the authors for their rebuttal. In general, I am pleased with all the improvements and clarifications that the authors have made. The rebuttal addresses most of my concerns. I am leaning towards acceptance of the paper since it maintains the high bar of the conference quality. I vote for 6.

---

### Official Review · Reviewer_wi3v · 2022-10-23

**Confidence:** 4
**Correctness:** 4
**Technical Novelty And Significance:** 2
**Empirical Novelty And Significance:** 2
**Recommendation:** 8

**Clarity, Quality, Novelty And Reproducibility:**

Overall, the paper is well-written and easy to follow. The evaluations are thorough. The proposed work combines several existing ideas -- using CLIP for zero-shot classification, task-specific prompt-tuning instead of fine-tuning, etc and shows good results on compositional zero-shot. I think the presented ideas are not novel by themselves, but its application for compositional zero-shot learning might be novel.

**Strength And Weaknesses:**


Strengths:
- The paper proposes a soft prompting approach for compositional zero-shot learning that improves performance with just a small set of learnable parameters (the tokens corresponding to attributes and classes).

- The paper show many useful experiments -- generalization to higher-order compositions, prompt tuning vs full fine-tuning, etc. Overall, the experiments are enough to demonstrate the effectiveness of the approach on the proposed task.

Weaknesses:
- Generally speaking, I am unsure if the evaluation is fair / comparable. The authors themselves point out that comparison to existing methods is difficult because they were not trained on large amounts of web data. The authors compare to CLIP and CoOp but the results here are not very surprising either -- task-specific learning should help improve performance.

- CoOp is a weird baseline to compare to. For CoOp, the prefix is learned, while for the proposed approach the token embeddings are learned. I think it will be nice to show if CSP + CoOp help over CSP. If you learn prefix tokens, as well as attribute and class tokens, does that help improve performance over just doing prefix learning / class & attribute token learning.

- Another baseline to compare to would have been learning token embeddings individually instead of together. So instead of using "A photo of [attribute] [class]", learn using "A photo of [attribute object", and "A photo of [class]". Can the model then combine these in interesting ways?

-- Finally, is it possible to avoid training at all on composition of attribute & classes at all. I think it's a little intellectually dissastisfying that despite training on so much web-scale data, we need to do more task-specific prompt tuning for recognizing compositions. Is there a smarter way to combine the two similarity scores (image-attribute, and image-class) that avoids the need to learn these embeddings?



**Summary Of The Paper:**

Summary:
The paper proposes a new method for the task of zero-shot compositionality on top of CLIP style models. The proposed approach learns embeddings for objects and attributes using the CLIP objective. At test time, they are able to recombine the embeddings for attributes and classes for novel compositions and show good zero-shot generalization.


**Summary Of The Review:**

Overall, the paper was easy to read and the presented idea was simple and intuitive. I am not sure that the paper in its current form presents any new significant insight that is not known to the broader AI community.
1. We know that CLIP are great at zero-shot tasks.
2. We know that prompt-tuning works really well instead of fine-tuning in a low-data regime.

I think the paper is missing a good-faith effort of getting CLIP to work for compositional zero-shot without any fine-tuning / prompt-tuning. It's a little dissatisfying that despite training on so much data, CLIP models need further task-specific training to perform well. Doing task-specific prompt-tuning seems like a very natural step, and I am unsure if the paper presents any ideas that will be useful to the broader AI community outside the niche of people who look at the compositional zero-shot learning task.

Update after rebuttal: I thank the authors for carefully addressing all the concerns in the rebuttal. I agree with the authors and reviewer  m1LH that the work is interesting in the context of compositional zero-shot learning. I am specially happy to see results for applying CLIP by learning the attribute embeddings without composition, and applying CLIP as it is. To reflect my updated thoughts on the paper, I am updating my score to 8

---

> ### Author Response · Authors · 2022-11-13
> **Response to Reviewer wi3v (1/2)**
>
> Thank you for your thoughtful comments on our manuscript. Below we address your comments and questions about our work.
>
> > Generally speaking, I am unsure if the evaluation is fair / comparable. The authors themselves point out that comparison to existing methods is difficult because they were not trained on large amounts of web data. The authors compare to CLIP and CoOp but the results here are not very surprising either -- task-specific learning should help improve performance.
>
> We would like to clarify that both CoOp and CSP are task-specific learning methods, but they differ in their style of prompting. What is perhaps surprising (or at least previously unstudied) is that by fine-tuning tokens in multiple compositions, even unseen compositions become significantly better without having to adjust the weights of the pre-trained model. Also, perhaps surprisingly, when fine-tuning data is limited, fine-tuning the entire pre-trained model doesn’t always improve performance over the pre-trained baseline, as we show in our experiments.
>
>
> > Another baseline to compare to would have been learning token embeddings individually instead of together. So instead of using "A photo of [attribute] [class]", learn using "A photo of [attribute object", and "A photo of [class]". Can the model then combine these in interesting ways?
>
> Thank you for your suggestion. We experiment with two variants of the prompt - A photo of [attribute] object and A photo of [object] where only the tokens in the square brackets are learnable parameters. We train the attribute soft prompts on attribute classification and object soft prompts on the object classification task. Then, during inference, we replace the CLIP token embeddings with the learned attribute and object embeddings. Our results on three datasets (see below) show that learning individual prompts can marginally improve or sometimes even hurt performance. This result shows that explicitly enforcing compositional learning of attributes and objects is key to the success of our method.
>
> |                      | MIT-States | UT-Zappos | C-GQA |
> |----------------------|------------|-----------|-------|
> | CLIP                 | 11.0       | 5.0       | 1.4   |
> | CSP (only attribute) | 11.7       | 6.2       | 2.0   |
> | CSP (only object)    | 8.9        | 7.0       | 1.5   |
> | CSP                  | 19.4       | 33.0      | 6.2   |
>
> > CoOp is a weird baseline to compare to. For CoOp, the prefix is learned, while for the proposed approach the token embeddings are learned. I think it will be nice to show if CSP + CoOp help over CSP. If you learn prefix tokens, as well as attribute and class tokens, does that help improve performance over just doing prefix learning / class & attribute token learning.”
>
> CoOp is one of the most popular styles of soft prompting in the vision-language community. In this work, we are proposing an alternative to this traditional way of soft prompting. It is, therefore, essential to compare CSP to CoOp. Further, we do not consider combining them because it is not clear to us how we could measure the benefit of compositional soft prompting relative to prefix prompting without running them separately.
>
>
> > Finally, is it possible to avoid training at all on composition of attribute & classes at all. I think it's a little intellectually dissastisfying that despite training on so much web-scale data, we need to do more task-specific prompt tuning for recognizing compositions. Is there a smarter way to combine the two similarity scores (image-attribute, and image-class) that avoids the need to learn these embeddings?
>
> We share a similar sentiment and find it dissatisfying that CLIP, out-of-the-box, does not work well on the compositional zero-shot learning task. We suspect that this is because contrastive pre-training on Web image captions does not require the model to learn precise compositional semantics. Our intellectual contribution is to demonstrate that a fine-tuning objective that explicitly enforces compositional semantics leads to better generalization without having to adjust the weights of the pre-trained model.
>
> As suggested, we try an additional experiment by combining the attribute and object prompts without fine-tuning any parameters. We take the attribute prompt and the object prompt from CLIP’s text encoder and average them together to get a single attribute-object representation. Our results below show that a naive combination of the attribute and object prompt hurts performance across all three datasets.
>
> |                          | MIT-States | UT-Zappos | C-GQA |
> |--------------------------|------------|-----------|-------|
> | CLIP                     | 11.0       | 5.0       | 1.4   |
> | CLIP (attr + obj prompt) | 6.9        | 1.8       | 0.7   |
> | CSP                      | 19.4       | 33.0      | 6.2   |

---

> > ### Author Response · Authors · 2022-11-13
> > **Response to Reviewer wi3v (2/2)**
> >
> >
> > > Overall, the paper was easy to read and the presented idea was simple and intuitive. .. Doing task-specific prompt-tuning seems like a very natural step, and I am unsure if the paper presents any ideas that will be useful to the broader AI community outside the niche of people who look at the compositional zero-shot learning task.
> >
> > As suggested by Reviewers 43mh and m1LH, prompting and compositionality are important areas of research in the machine learning community. Besides the benchmark results, we show the benefits of CSP in generalizing to higher-order compositions and combinations of pretrained attributes and fine-tuned objects. These evaluations have not been considered in previous works.
> >
> > Furthermore, large-scale pre-trained models are increasingly common, and enhancing capabilities like compositionality is valuable to the community. As suggested by Reviewer m1LH, our work can serve as a “strong baseline to build upon for future work”. Further, we believe there are many domains where learning specialized, composable vocabularies on top of pre-trained models will be valuable.

---

### Official Review · Reviewer_1rcT · 2022-10-27

**Confidence:** 4
**Correctness:** 3
**Technical Novelty And Significance:** 2
**Empirical Novelty And Significance:** 3
**Recommendation:** 5

**Clarity, Quality, Novelty And Reproducibility:**

As noted above, the paper is clear, yet not so strong in terms of originality as it appears to be mainly an adaptation of a simple soft-prompting approach to the compositional zero-shot learning problem.

A question regarding the clarity: do we expect the model to improve the unseen attribute and unseen class based cases at all?  (I guess not, unless the combination includes a seen attribute or a seen object, right?)

**Strength And Weaknesses:**

Strengths:
- The paper presents a rather comprehensive experimental comparison with favorable results. The presented simple approach appears to be effective and overall quite well analyzed in terms of benchmark-based experimental comparisons.
- The clarity of model selection details is a plus.


Weaknesses:
- The paper appears to be mainly an adaptation of a simple soft-prompting approach to the compositional zero-shot learning case.
- As noted in the paper,  the underlying CLIP model is trained on a very large training set, even if that's possibly noisy. While CLIP has been used in various works for "zero-shot classification", it is hardly zero-shot; therefore, the problem can instead be seen synthesizing MxN-classifiers out of CLIP.  To this end, I find the criticism of "task-specific architures" in terms of adaptability and parameter complexity, at the end of Section 2 misleading and imprecise.
- The model selection discussion emphasizes that all hyper-parameters except the unseen bias are tuned on the validation set wrt unseen performance. Here, I question the validity of tuning the bias term directly on the test set, which breaks the zero-shot setting, even if some prior work has taken this approach. What happens if you were to tune the bias term itself on the validation set, instead of the test set? How much performance drop would it cause?


**Summary Of The Paper:**

The paper tackles the compositional zero-shot learning through the use of a large-scale pretrained vision-language model, specifically: CLIP. The paper takes the prompt-based classification approach where the cosine similarity between the text and vision embedding is used as the compatibility score. To improve this scheme, the paper takes a soft-prompt approach by fine tuning the prompt tokens' text embeddings. The fine-tuned object/attribute embeddings are then used in prompt-based zero-shot classification. T

**Summary Of The Review:**

The paper is a good, mainly-experimental work on the soft-prompting based use of CLIP for compositional classification. The paper is clear and contains good experimental analysis. The technical novelty appears to be rather weak, the work is mainly a simple engineering solution (though I definitely do not question its value). I have concerns regarding the bias-term selection policy.

---

> ### Author Response · Authors · 2022-11-13
> **Response to Reviewer 1rcT**
>
> Thank you for your thoughtful comments on our manuscript. Below we address your comments and questions about our work.
>
> > The paper appears to be mainly an adaptation of a simple soft-prompting approach to the compositional zero-shot learning case.
>
> We respectfully disagree on this point. We believe that CSP’s novel approach of tuning many tokens in multiple compositions and then testing on unseen compositions is a significant advance in soft prompting. Soft prompting has become popular in part due to its simplicity. To the best of our knowledge, across all this work, CSP is the first method that aims to learn prompt tokens with specific meanings that are finer-grained than classes. Our experiments show that the CSP approach leads to significantly improved performance. Further, the more compositions that are seen training, the better the tuned tokens compose, even with pre-trained vocabulary. This result demonstrates that the CSP learning objective is achieving its intended effect, which is the technical contribution of the paper.
>
> > As noted in the paper, the underlying CLIP model is trained on a very large training set, even if that's possibly noisy. While CLIP has been used in various works for "zero-shot classification", it is hardly zero-shot; therefore, the problem can instead be seen synthesizing MxN-classifiers out of CLIP. To this end, I find the criticism of "task-specific architures" in terms of adaptability and parameter complexity, at the end of Section 2 misleading and imprecise.
>
> Yes, we completely agree that the popularity of CLIP-like models has diluted the meaning of the term “zero-shot.” We think it is important to study how to improve these models since they are now central to many state-of-the-art machine learning approaches. As noted in Section 2, this leads to the same caveat as in the original CLIP work and all subsequent adaptations: we don’t know exactly what it was pre-trained on. We, therefore, emphasize the comparisons with other CLIP-based methods that all have the same pre-training.
>
> We’re concerned that we don’t fully understand your point about why this means our discussion of task-specific architectures is “misleading” and “imprecise.” As we say, from the perspective of an end-user, prompt-tuning requires tuning orders of magnitude fewer parameters. We do not factor the many more parameters in CLIP that were learned during pre-training into the comparison because a user is faced with the choice of prompt tuning or learning a task-specific architecture. As our experiments show, CLIP-based methods are also more flexible because we can adapt to things like attribute-attribute-object classification and mixtures of fine-tuned and pre-trained vocabulary. We would be grateful for clarification and can update the discussion based on any specific suggestions you have.
>
>
>
> > The model selection discussion emphasizes that all hyper-parameters except the unseen bias are tuned on the validation set wrt unseen performance. Here, I question the validity of tuning the bias term directly on the test set, which breaks the zero-shot setting, even if some prior work has taken this approach. What happens if you were to tune the bias term itself on the validation set, instead of the test set? How much performance drop would it cause?
>
> We would like to clarify that a bias term is only tuned on the test set for some of the evaluation metrics (best seen, best unseen, and best harmonic mean), which we report for consistency with the literature. The AUC metric does not tune the bias, instead measuring the quality of the tradeoff offered between seen and unseen accuracy via the area under the curve induced by all possible bias terms. This metric is therefore not affected by information leaking. This is why we emphasize the AUC metric when discussing our results in the abstract and introduction, as well as the results reported in Tables 3 and 4, and Figure 5.

---

### Author Response · Authors · 2022-11-13
**To all reviewers**

We thank all the reviewers for your thoughtful comments and feedback on our manuscript. We are very grateful for the time you have put into giving us feedback. In addition to individual replies, here we summarize the main clarifications of our manuscript.

### Technical Novelty

Reviewers 1rcT, wi3v, and 43mh have questioned the technical novelty of compositional soft prompting. We would like to clarify that even though CSP is simple to implement, it has several important implications for prompting and compositionality. We are the first to show that explicitly enforcing compositionality when fine-tuning soft prompts can lead to more composable vocabulary and significantly improve the performance of CLIP and traditional soft-prompt tuning. Prior works, like CoOp and visual prompt tuning, treat prompts as fixed inputs that do not change between training and testing. The CSP approach of tuning many compositions of tokens and then generalizing to new compositions is a fundamentally new way of tuning prompts. We are excited about the potential significance of this work, as we believe there are many application areas that could benefit from learning specialized, composable vocabulary.

Finally, as suggested by Reviewer m1LH, our findings are of significant interest to researchers studying compositionality. Large-scale pre-trained models are becoming increasingly common, and enhancing capabilities like compositionality is valuable to the community.

### Additional Experiments and Alternative Comparisons

Reviewers wi3v, 43mh, and m1LH suggested alternative prompting comparisons, including visual prompt tuning, to further investigate the benefits of our work. To that end, we perform additional experiments and show that compositional soft prompting significantly outperforms these methods. For more details on these experiments and results, see the individual replies to the reviewers.

---

### Decision · Program_Chairs · 2023-01-20

**Decision:**

Accept: poster

**Justification For Why Not Higher Score:**

Several reviewers find the novelty sufficient but not overwhelming.

**Justification For Why Not Lower Score:**

No reviewer recommends rejection.

**Metareview: Summary, Strengths And Weaknesses:**

This paper describes a way to learn attribute+object prompts to be used in unseen combinations. The reviewers generally appreciate the idea, although some find it is not completely novel. They appreciate the experiments on a number of datasets, but question some aspects of the setup (choice of methods compared, use of test set). The rebuttal was effective overall, so most reviewers lean positive, and two confidently recommend acceptance.

**Note From Pc:**

if the above contains the word "oral" or "spotlight" please see: "oral" presentation means -> notable-top-5% and "spotlight" means -> notable-top-25%. As stated in our emails, we are disassociating presentation type from AC recommendations